# Methods to Isolate Muscle Stem Cells for Cell-Based Cultured Meat Production: A Review

**DOI:** 10.3390/ani14050819

**Published:** 2024-03-06

**Authors:** Jae-Hoon Lee, Tae-Kyung Kim, Min-Cheol Kang, Minkyung Park, Yun-Sang Choi

**Affiliations:** Research Group of Food Processing, Korea Food Research Institute, Wanju 55365, Republic of Korea; l.jaehoon@kfri.re.kr (J.-H.L.); ktk.kim@kfri.re.kr (T.-K.K.); mckang@kfri.re.kr (M.-C.K.); mk.park@kfri.re.kr (M.P.)

**Keywords:** muscle stem cells, cell isolation, pre-plating, density gradient centrifugation, fluorescence- and magnetic-activated cell sorting

## Abstract

**Simple Summary:**

Cultured meat development relies on diverse cell types, with muscle stem cells (MuSCs) standing out for their superior muscle differentiation abilities. Extracting MuSCs from muscles involves methods like pre-plating based on cell adhesion, density gradient centrifugation using cell density variations, and fluorescence- and magnetic-activated cell sorting relying on molecular markers. As the cultured meat industry progresses, the demand for more efficient separation techniques is expected to grow. This review delves into these existing methods and explores future avenues for isolating MuSCs for cultured meat production.

**Abstract:**

Cultured meat production relies on various cell types, including muscle stem cells (MuSCs), embryonic stem cell lines, induced pluripotent cell lines, and naturally immortalized cell lines. MuSCs possess superior muscle differentiation capabilities compared to the other three cell lines, making them key for cultured meat development. Therefore, to produce cultured meat using MuSCs, they must first be effectively separated from muscles. At present, the methods used to isolate MuSCs from muscles include (1) the pre-plating method, using the ability of cells to adhere differently, which is a biological characteristic of MuSCs; (2) the density gradient centrifugation method, using the intrinsic density difference of cells, which is a physical characteristic of MuSCs; and (3) fluorescence- and magnetic-activated cell sorting methods, using the surface marker protein on the cell surface of MuSCs, which is a molecular characteristic of MuSCs. Further efficient and valuable methods for separating MuSCs are expected to be required as the cell-based cultured meat industry develops. Thus, we take a closer look at the four methods currently in use and discuss future development directions in this review.

## 1. Introduction

Cell-based cultured meat, a cutting-edge food technology, involves artificially producing meat products from animal cells [1], offering high-protein, high-quality food without the need for animal slaughter. It considerably differs from traditional livestock farming which involves raising and slaughtering livestock [2]. This technology is anticipated to yield environmental, ethical, and health benefits [3], garnering attention for its potential to help achieve carbon neutrality and ensure food security [4]. Despite this, commercial production remains limited, due to unresolved technical (difficulty in mass culturing), ethical (difficulty replacing fetal bovine serum), and regulatory (absence of relevant laws and regulations in each country) challenges, which are expected to be resolved through ongoing research and technological advancements.

Using muscle stem cells (MuSCs) for cultured meat production offers numerous benefits. MuSCs enable the precise formation of the desired muscle tissue, enhancing the meat quality with a delicate and consistent texture and taste [5]. These properties promote rapid cell proliferation, improving production efficiency [6]. Additionally, MuSCs allow for the control over muscle tissue characteristics [5], facilitating customization of nutritional and fat contents, as well as taste, to meet consumer preferences [7]. These advantages make cultured meat a sustainable and environmentally friendly alternative. In addition, cultured meat using MuSCs is expected to increase its portion in the global meat market as a new protein source. According to a 2019 report, 35% of the entire meat market will be replaced by cultured meat by 2040 [8]. To this end, many related companies are investing in large-scale production facilities for industrial-scale production.

Effective MuSC isolation from various livestock species is crucial for optimizing the culture process and ensuring cells grow and differentiate under the desired conditions [9]. Cell separation is essential for creating an optimal culture environment for a desired cell line or type and obtaining cells with desired characteristics [10]. Isolated cells rapidly proliferate in suitable culture media and conditions, leading to increased cell quantities, an improved productivity [11], and an improved quality of the final cultured meat product. Various methods like pre-plating, density gradient centrifugation, fluorescence-activated cell sorting (FACS), and magnetic-activated cell sorting (MACS) utilize the physical, biological, and molecular features of MuSCs to isolate them from cell-based cultured meat.

Therefore, this study aimed to describe and utilize the methods of MuSC isolation to effectively isolate MuSCs to produce cell-based cultured meat.

## 2. MuSC Dissociation from Tissue

Cells existing within the muscle must be separated from muscle tissue to isolate MuSCs. Research in the 1960s revealed that skeletal MuSCs are located beneath the basal lamina of the skeletal muscle fibers [12]. Processes involving decomposition of muscle tissue through physical treatment, dissociation of the cells present within the muscle through chemical treatment, and removal of non-cellular debris (fiber fragments, tissue debris, and connective tissues) through filtration or differential centrifugation are required to effectively isolate MuSCs [13].

Using sterilized forceps and scissors, fat, connective tissue, bone, and tendons are first removed from the muscle tissue. At this time, surgical scissors or a scalpel may be used, as well as a small grinder to grind the meat [14,15]. At this stage, preventing contamination from microorganisms is crucial, and the meat should be finely ground for proteolytic enzyme action in the next step to be effective. The subsequent step involves separating the muscle-derived cells from the muscle tissue using various proteolytic enzymes. Examples of commonly used enzymes include trypsin, pronase, collagenase, and dispase [14,16,17]. Studies have reported cell dissociation using the above enzymes alone [18,19]; however, studies have reported the use of various enzyme combinations to increase the dissociation efficiency given that each enzyme has a different substrate specificity. When using enzyme combinations, cases exist where they are sequentially used to provide the optimal reaction conditions for each enzyme [17,20]. However, cases where all enzymes are simultaneously applied to reduce the reaction time from an efficiency perspective have also been reported [21,22]. Therefore, the efficiency of isolating MuSCs can be increased by determining the optimal proteolytic enzyme conditions for each condition, including the livestock species and muscle types. Finally, several methods are used to remove the noncellular debris generated during the previous digestion process. Only the cells are generally separated by filtration using a cell strainer or nylon mesh with a small pore size because the weight and size of non-cellular debris are larger than those of cells. Pore sizes of 40, 70, and 100 μm are mainly used, and while some studies have conducted separation using a single pore size [23,24], studies have also reported an increase in efficiency by sequentially filtering starting from larger sizes [13,25]. Previous studies have reported cell separation using differential centrifugation without filtering [26]. Additionally, studies have reported that adding a process using a certain number of needles increases the dissociation efficiency of the cell suspension [22,27].

## 3. MuSC Isolation Methods

Various cell types exist in the cell mixture after separating muscle cells, including somatic cells, blood cells, stromal cells, epithelial cells, fibroblasts, and MuSCs. Therefore, to isolate MuSCs, various methods have been developed [28,29]. Pre-plating, density gradient centrifugation, FACS, and MACS are widely employed to isolate MuSCs from various species, including mice, rats, pigs, cattle, goats, chickens, and humans. Each isolation method has distinct principles, advantages, and disadvantages; therefore, they are selected and used depending on the research needs (Table 1).

Firstly, the pre-plating method leverages MuSCs’ distinct adhesion ability compared to other cells like fibroblasts and epithelial cells [29]. MuSCs have a lower ability to adhere to the surface compared to other cells due to their lower adhesion ability, making them suitable for separation using the pre-plating methods. This method offers simplicity and a relatively quick separation process as the main advantages [29]. Additionally, this method has an economic advantage in that it does not require expensive equipment or reagents [30]. However, this process may take a long time depending on the separation conditions [25], and the purity of the isolated MuSCs may be low owing to unintended cell attachment [31].

Secondly, the density gradient centrifugation method, which uses MuSCs’ physical characteristics, is a separation method that uses the difference in the intrinsic cell density between MuSCs and other cells [32]. This method creates a density gradient using a dedicated substrate and separates cells based on the differences in cell density through centrifugation. This process has some advantages in that it is simple, the time required for the process is short, and the process is economical. Additionally, further precise cell separation is possible by customizing the substrate concentration [33]. However, this process provides low yield and purity and is difficult for MuSCs to recover from the fraction during separation [34].

Thirdly, FACS and MACS isolation methods utilize the molecular characteristics of MuSCs such as the specific protein markers on the cell surface unlike other cells [24,35]. The common advantage of both separation methods is that they can isolate high-purity MuSCs via antigen–antibody reactions [35]. Additionally, in FACS, separation is possible based on surface markers, cell size, and granularity [24]. MACS is advantageous in that it has a relatively short separation time compared to FACS and is easy to scale up for industrialization [36]. However, the disadvantages of both separation methods include the need for additional equipment and reagents to separate MuSCs and the potential for damage to the isolated cells due to electric and magnetic fields [37,38].

Next, we will examine the detailed separation method for each technique and explore various application methods based on actual cases.

### 3.1. Pre-Plating Method

Pre-plating is an isolation method that exploits the difference in adhesion ability between MuSCs and other cells (epithelial cells, fibroblasts, and stromal cells) [29]. This method involves culturing the muscle cell mixture obtained through dissociation and subsequently culturing the non-adherent MuSCs from the supernatant to a new flask after a certain period (Table 2).

Pre-plating typically lasts 1–2 h and uses non-coated cell culture dishes or flasks for primary culture to emphasize the differences in cell adhesion [18,21,47]. Li and Wang [30] separated MuSCs from pigs under varied conditions with 0.5, 1.5, and 3 h pre-plating times and compared and analyzed the cell number, PAX7^+^ cell proportion, and differentiation capacity under each condition. The cell number was the highest in the cells pre-plated for 0.5 h, and the PAX7^+^ cell proportion was similar to that of the other groups. Additionally, the differentiation capacity confirmed after 20 d was the highest for the cells pre-plated for 0.5 h and this was five times higher than that of the FACS-separated cells used as a control group. Thus, setting a pre-plating time for each study was necessary to increase the efficiency of MuSC isolation.

Various studies have been conducted to improve the purity and efficiency of isolating MuSCs by repeating pre-plating multiple times. The pre-plating process can be attempted two to seven times, and MuSCs can be isolated under various pre-plating conditions (1, 2, 24, and 48 h) [41,42,43]. Various research teams have performed pre-plating at 24 h intervals for up to 72 h to isolate MuSCs from mice [15,37,39]. These studies reported that cells from the last fraction had a high percentage of desmin-positive cells, making this fraction suitable for future research on MuSCs.

However, long-term pre-plating does not necessarily increase the MuSC purity [25,40,45]. Nolazco et al. [40] isolated MuSCs from mouse hind limb muscles by pre-plating under various conditions (pre-plating every 24 h until day 6). They confirmed that the third and fourth fractions demonstrated the highest satellite cell proportion. Additionally, Redshaw et al. [25,45] performed pre-plating every 24 h until the seventh day when isolating MuSCs from pigs and reported that the myotube-generating activity was satisfactory in the fraction from 2 h to 48 h.

Recently, studies on adding a shaking process to increase the efficiency of pre-plating have been reported [29,46]. Yoshioka et al. [29] isolated MuSCs from human and mouse muscles using a pre-plating method with incubation and shaking steps added (shaking the dish every 5 min 10 times). MuSCs, which have a lower adhesion ability than fibroblasts, are challenged by the shaking step to attach to the culture dish surface. Therefore, this maximizes the separation effect. Kim et al. [46] studied the effects of shaking on the isolation of MuSCs from chickens. They analyzed the characteristics of MuSCs isolated by varying the shaking time (shaking every 8 or 15 min) and pre-plating time (1, 2, 3, and 4 h). The PAX7^+^ cell proportion was the highest when the first and second pre-platings were performed for 2 h and 2 h with shaking (shaking the dish every 8 min), respectively.

### 3.2. Density Gradient Centrifugation

Gradient centrifugation sorts cells based on their size or density [32] and using the relative difference in the inherent cell density [48]. A density gradient can be created using a density gradient medium such as Percoll, and the cells are isolated via centrifugation (Table 3).

In a pig study, dissociated cells were isolated via 20% or 20%/60% Percoll discontinuous gradient centrifugation [54,55]. Perruchot et al. [54] considered the cells in the lower 20% density to be pig satellite cells, and the cell yield was confirmed to be 1.5 × 10^6^ cells/g of muscle. They conducted a proliferation test using the pig satellite cells for 7 d and confirmed that the percentage of desmin-positive cells was >96%. In some studies, the cell subpopulation was further divided into three density concentrations [19,50,53,56,57]. Thus, these studies attempted to increase the purity and efficiency of MuSC isolation. Mau and Oksbjerg [53] isolated MuSCs at densities of 25%, 40%, and 90%, and a 40/90% interface fraction was used in the experiment. The MuSC yield was 4.1 × 10^6^ cells/g of muscle, and the ratio of desmin-positive cells was reported to be >95%.

Miersch et al. [19] isolated MuSCs from pigs via density gradient centrifugation, separated them under different density conditions (condition 1: 25%, 40%, and 70%; condition 2: 25%, 40%, 50%, and 70%), and their myogenic activity was compared and analyzed. In their study, a 40/70% interface fraction was obtained under condition 1, which was divided into 40/50 and 50/70% subpopulations under condition 2. The 40/50% fraction of cells obtained by subdividing and adding one density concentration revealed a faster proliferation activity, a higher fusion rate, and medium- and large-sized myotube formation, confirming their high myogenic differentiation potential.

However, despite the cells being separated at various densities, low yields have also been reported. Che et al. [33] and Kästner et al. [49] isolated MuSCs from rat muscles using density gradient centrifugation. Che et al. [33] used six densities (15%, 25%, 35%, 40%, 55%, and 70%) and separated them into five fractions. The 15/25% interface fraction had the highest proportion of stem cell antigen-1 (Sca-1) cells, which was used as a stem cell marker, but the value did not exceed 70%. Additionally, Kästner et al. [49] obtained seven fractions via density gradient centrifugation (five density concentrations were used), of which the fraction with the highest proportion of myogenic cells contained 52.6% of the myogenic cells.

### 3.3. Fluorescence-Activated Cell Sorting (FACS)

FACS involves isolating MuSCs using the specific surface markers possessed by these cells. Surface markers are proteins that are highly expressed in specific cells compared to other cells, and sorting MuSCs using antibodies that bind to these proteins is possible [24]. Only one marker was used to separate the cells in the early stages of MuSC research using FACS. However, various markers were developed to increase the purity and efficiency, and research was conducted to distinguish between negative and positive markers [31]. We reviewed the related studies that focused on this point (Table 4).

Several studies have reported the isolation of MuSCs from various species ranging from mice to pigs, cattle, and humans using FACS, which is performed using the specific surface markers expressed in MuSCs for each species. In mouse studies, integrin α7 and CD34 are commonly used as positive markers [58,60,61]. Additionally, Fu et al. [58] and Pasut et al. [61] attempted to increase the MuSC purity using CD31, CD45, CD11, and Sca-1 as the negative markers. Consequently, the ratio of PAX7-expressing cells in the integrin α7^+^CD34^+^/CD31^−^CD45^−^CD11^−^Sca-1^−^ sorted cells was >90% [61]. Maesner et al. [60] performed FACS by adding integrin α7 and CD34, as well as calcein, CD29, CXCR4, and VCam1, as the positive surface markers. MuSCs were separated by dividing the combination of positive markers into three (fraction 1; Calcein^+^CD29^+^CXCR4^+^, fraction 2; Calcein^+^VCam1^+^, fraction 3; Calcein^+^Integrin α7^+^CD34^+^) when sorting cells in this study, and experiments were conducted to confirm the ratio of PAX7-expressing cells. Thus, setting the surface markers is extremely crucial when sorting cells using FACS. Urbani et al. [59] used the SM/C2.6 positive marker to isolate MuSCs from mice. The cell proportions expressing PAX7 and integrin α7 in MuSCs isolated using the SM/C2.6 positive marker were 94% and 96%, respectively. Moreover, SM/C2.6, a novel monoclonal antibody against mouse muscle satellite cells, was developed as a cell surface marker for quiescent satellite cells in murine skeletal muscles [65].

In pig and cattle studies, the same cell surface markers are used; positive markers include CD29 and CD56, whereas the negative markers are CD31 and CD45 [10,30,44,62,63]. These surface markers can be used in both species, and the ratio of PAX7-expressing cells in MuSCs obtained using these markers is high (Ding et al. [44]: 94%; Li et al. [30]: 95%; Zhu et al. [62]: 87%; Ding et al. [63]: 92%). Particularly, the proportions of cells expressing MYOD, desmin, and MYF5 were 98.48%, 98.57%, and 97.01%, respectively, when MuSCs were isolated from cattle as CD56^+^CD29^+^/CD31^−^CD45^−^ [63].

In a human study, ITGB1 was used as a positive marker, whereas CD31, CD34, and CD45 were used as negative markers [64]. ITGB1, not used in mice, pigs, or cattle, was chosen as a positive marker, while CD34, previously a positive marker in mice, was used as a negative marker in human studies. This underscores the significance of identifying species-specific optimal surface markers.

### 3.4. MACS

MACS operates on a principle similar to FACS, sorting MuSCs based on specific surface markers, but utilizes a magnetic field instead of fluorescence for sorting [35]. Consequently, a secondary antibody conjugated to microbeads is employed rather than a fluorescent substance (Table 5).

In studies involving pigs and cattle, CD29 was used as a positive marker for cell sorting similar to FACS [14,67]. In these studies, muscle cells were dissociated from the muscle tissue and reacted with an anti-CD29 antibody and anti-mouse IgG microbeads as the primary and secondary antibodies, respectively. A dedicated column was mounted on the MACS separator with a strong magnetic field, and the cell mixture containing the magnetic microbeads was flowed to allow the cells that were not labeled with CD29 to flow down. The MuSCs were separated from the column using a flowing buffer after detaching the column from the MACS separator. Choi et al. [14] reported that the proportion of the myogenic cells could be increased to 91.5% using MACS despite the proportion before MACS being 30%.

In a human study, MuSCs were isolated using the aforementioned method, and CD56 was used as a positive marker [68,69,70,71]. This marker is one of the positive markers used to separate MuSCs from pigs and cattle in cell separation using FACS. Thus, the similarity between the surface markers of pigs, cattle, and humans is confirmed. MuSCs with a high purity and yield can be obtained through MACS separation using CD56.

In a mouse study, CD44, CD90, and integrin α7 were simultaneously used as positive surface markers for MuSC isolation [23]. Elashry et al. [23] explored various positive markers to enhance the efficiency of MuSC isolation. Studies using negative markers have also reported an increased MuSC separation efficiency [16,66]. Sincennes et al. [16] reported the reaction of the negative markers CD31, CD45, Sca1, and CD11b with the cell mixture and reaction of secondary antibodies with the microbeads. Subsequently, cells that did not bind to the negative marker were obtained using an MACS separator. Finally, the obtained cells reacted with the positive marker integrin α7 and the microbeads, and integrin α7^+^ cells were obtained using an MACS separator. Through this, contamination of unwanted cells such as fibroblasts was reported to be more effectively reduced when separating MuSCs.

## 4. Research to Increase MuSC Separation Efficiency

Research has aimed to enhance the MuSC purity and yield during separation using the described methods. Optimization strategies include setting the time, conditions, and frequency of pre-plating; setting the density concentration in gradient density centrifugation; setting appropriate surface markers; and combining the use of positive and negative markers in FACS and MACS.

However, research has been conducted to increase MuSC purity using a combination of several cell isolation methods in addition to the aforementioned methods. Tripathi et al. [57] isolated MuSCs from goats using density gradient centrifugation (20%, 40%, and 90%). They used a 20/40% interface fraction for the MuSCs. Notably, a pre-plating process was performed before density gradient centrifugation in this study. The authors were able to remove a substantial portion of fibroblasts and perform density gradient centrifugation after dissociating the cells from the muscle, digesting them with enzymes and putting them through a 3 h pre-plating process. Additionally, the authors reported that they were able to obtain MuSCs with a purity of over 90% through a combination of pre-plating and density gradient centrifugation (pre-plating only: 65–70%, density gradient centrifugation only: 75–80%). Bareja et al. [72] efficiently isolated MuSCs from humans using MACS and FACS simultaneously. MACS was performed using CD11b, CD31, CD34, and CD45 as the negative markers for primary cell sorting. Subsequently, MuSCs were obtained by performing secondary cell sorting using FACS with CXCR4 and CD56 as positive markers.

Benedetti et al. [73] introduced an ice-cold treatment (ICT) method to efficiently isolate MuSCs, which involves briefly incubating the cell culture dish on ice (0 °C) for 15–30 min. Thus, only MuSCs are separated from the culture dish, and when compared with other commonly used separation methods, this method increases the purity. The authors reported that the myogenic cell ratio was significantly higher when compared to MuSCs derived from the ICT method and MuSCs isolated using MACS. Consequently, this new method has several advantages, including cost-effectiveness, accessibility, and technical simplicity. Future research should continue to improve the efficiency of MuSC isolation to develop the cell culture food industry.

## 5. Conclusions and Future Directions/Recommendations

We reviewed four key methods for MuSC isolation: pre-plating, density gradient centrifugation, FACS, and MACS. These methods differentiate MuSCs based on their physical, biological, and molecular characteristics. Given the advantages and disadvantages of each method, a complementary approach is favored for MuSC isolation. Additionally, optimizing the conditions can enhance the purity and yield, regardless of the method used, depending on the livestock type and muscle source from which the MuSCs are to be separated. Therefore, selecting an optimal separation method suitable for the research conditions and conducting optimization studies are considered necessary for each study. Although MuSCs have the disadvantage of a decreased proliferation capacity and stemness as they are subcultured, it is true that they hold an advantageous position in the production of cell-cultured meat due to their significantly higher differentiation ability compared to other cell types (embryonic stem cell lines, induced pluripotent cell lines, and naturally immortalized cell lines). As the industry develops, cultured meat from MuSCs is anticipated to rival real meat in premium markets, emphasizing the importance of continued research into efficient MuSC isolation. Additionally, combining separation methods with new technologies is vital to further enhance the efficiency.

## Figures and Tables

**Table 1 animals-14-00819-t001:** Comparison of methods to isolate MuSCs.

	Methods	Pre-Plating	Density Gradient Centrifugation	FACS	MACS
Trait					
Principle		- Uses biological features of MuSCs - Utilizes the fact that adhesion ability is lower compared to other cells	- Uses physical features of MuSCs - Utilizes the intrinsic cell density of MuSCs that is different from other cells	- Uses molecular features of MuSCs - Utilizes the unique cell surface protein marker of MuSCs	- Uses molecular features of MuSCs - Utilizes the unique cell surface protein marker of MuSCs
Advantages		- The process is simple - The time required for the process is short - It is economical as it does not require expensive equipment or reagents	- The process is simple - The time required for the process is short - It is economical as it does not require expensive equipment or reagents - Customization of the concentration used is possible	- High-purity MuSCs can be obtained through antigen–antibody reactions - In addition to surface markers, sorting by cell size and granularity is possible	- The separation time is shorter compared to FACS - High-purity MuSCs can be obtained through antigen–antibody reactions - A high possibility exists that it can be scaled up for industrial use
Disadvantages		- May take a long time depending on the separation conditions - MuSCs may be lost due to unintentional attachment to the surface of the culture dish during pre-plating	- Low separation efficiency and purity - Difficulty in isolating MuSCs in the fraction	- Requires expensive equipment (flow cytometry) and reagents - Maintenance costs are high - Electric charges applied to cells during sorting can cause gene expression changes	- MACS separator and dedicated column are required - Magnetic properties can damage the cell membranes

MuSCs, muscle stem cells; FACS, fluorescence-activated cell sorting; MACS, magnetic-activated cell sorting.

**Table 2 animals-14-00819-t002:** MuSC isolation using the pre-plating method.

Species (Muscle Types)	Detailed Methods	Specific Features	References
Mouse (limb muscles)	- Cells were cultured on a 10 cm cell culture dish for 1.5 h. - The non-adherent cells were collected and cultured onto a 6-well plate.	-	Contreras, Villarreal [21]
Mouse (hind limb muscles)	- Cells were cultured on a collagen-coated flask for 1 h. - The non-adherent cells were transferred and cultured for 2 h. - The non-adherent cells were transferred and cultured for 21 h. - Subsequently, pre-plating was performed every 24 h for up to 72 h.	- Pre-plating was performed 6 times at different times (1, 3, 24, 48, 72, and 96 h). - The percentage of desmin-positive cells was the highest in the 6th pre-plating cell at 94.6%.	Jankowski, Haluszczak [37]
Mouse (hind limb and gastrocnemius muscle)	- Cells were cultured on a collagen-coated flask for 2 h. - The non-adherent cells were transferred and cultured for 18 h. - Subsequently, pre-plating was performed every 24 h for up to 72 h.	- Pre-plating was performed 5 times at different times (2, 20, 44, 68, and 92 h). - Cells that survive the last pre-plating are used as the MuSCs.	Lavasani, Lu [15]
Mouse (hind limb muscles)	- Cells were cultured on a collagen-coated flask for 1 h. - The non-adherent cells were transferred and cultured for 2 h. - The non-adherent cells were transferred and cultured for 18 h. - Subsequently, pre-plating is performed every 24 h for up to 72 h.	- Pre-plating was performed 6 times at different times (1, 3, 21, 45, 69, and 93 h). - The percentage of desmin-positive cells was the highest in the 6th pre-plating cell at 78%.	Qu, Balkir [39]
Mouse (hind limb muscles)	- Cells were cultured on a collagen-coated flask for 2 h. - The non-adherent cells were transferred and cultured in a collagen-coated flask every 24 h for 6 d.	- Pre-plating was performed 6 times every 24 h. - The flask on the 2nd day had a high fibroblast proportion, and the satellite cell proportion was high on the 3rd and 4th days.	Nolazco, Kovanecz [40]
Rat (limb muscles)	- Cells were cultured on a collagen-coated dish for 2 h. - The non-adherent cells were transferred to a new collagen-coated dish and incubated for 2 d. - The cells were detached and a second pre-incubation was conducted for 1 h. - The second non-adherent cells were collected and cultured on a collagen-coated dish.	- Pre-plating was performed twice, and the purity of satellite cells was increased to >90%.	Dai, Yu [41]
Rat (hind limb muscles)	- Cells were cultured on an uncoated dish for 2 h. - The non-adherent cells were transferred and cultured for 24 h. - The non-adherent cells were transferred onto a collagen-coated dish and cultured.	-	Machida, Spangenburg [42]
Rat (gastrocnemius muscles)	- Cells were cultured on an uncoated 150 mm dish for 3 h. - The non-adherent cells were transferred and cultured for 3 h. - This pre-plating was repeated two more times.	- Pre-plating was performed four times for 3 h each.	Chakravarthy, Davis [43]
Pig (NS ^1^)	- Cells were cultured on the uncoated dish for 1 h. - The non-adherent cells were collected and cultured on the collagen-coated dish for 2 d. - The cells were detached and this pre-plating was repeated two more times.	- Pre-plating was performed three times.	Ding, Wang [44]
Pig (diaphragm, hind limb muscles)	- Cells were initially cultured on plastic for 2 h. - The non-adherent cells were transferred and cultured in collagen-coated flasks every 24 h for 7 d.	- Pre-plating is performed seven times every 24 h. - Cells between 2 and 48 h were confirmed to have good myotube-generating activity.	Redshaw, McOrist [25], Redshaw and Loughna [45]
Pig (hind limb muscles)	- Cells were cultured on a flask for 1 h. - The non-adherent cells were collected and cultured.	-	Li, Li [38]
Pig (NS)	- Cells were cultured on a collagen-coated plate for different times (0.5, 1.5, and 3 h). - The non-adherent cells were transferred onto a matrigel-coated plate and cultured.	- When comparing the expansion and differentiation efficacy of cells separated using FACS and MuSCs, 0.5 h pre-plating cells were superior.	Li, Wang [30]
Chicken (embryos)	- Cells were cultured on a collagen-coated flask for different times (2, 3, and 4 h). Subsequently, the non-adherent cells were transferred and cultured for 2 h at 37 °C with shaking every 15 min. The non-adherent cells were transferred to a new flask and cultured. - Cells were cultured on a collagen-coated flask for different times (1, 2, and 3 h). Subsequently, the non-adherent cells were transferred and cultured for 2 h at 41 °C with shaking every 8 min. The non-adherent cells were transferred to a new flask and cultured.	- Pre-plating is performed with different incubation temperatures and shaking times. - As a result of examining pax7-positive cell content, pre-plating at 41 °C for a total of 4 h was the most efficient method.	Kim, Kim [46]
Chicken (embryo)	- Cells were cultured on a flask for 1 h. - The non-adherent cells were collected and cultured.	-	Ryu, Kim [18]
Human (semitendinosus muscles), Mouse (tibialis anterior muscles)	- Cells were cultured on a collagen-coated dish for 16 h. - The non-adherent cells were transferred and cultured for 3 h. - The non-adherent cells were transferred and cultured for 24 h. - After 24 h, the media was replaced with fresh media and cultured for an additional 24 h. - The attached cells were detached and transferred to a collagen-coated dish. At this step, the dish was incubated for 5 min and then gently shaken. - After repeating this 5 times, non-adherent cells were transferred to a new matrigel-coated dish and cultured.	- During the pre-plating process, a shaking step was added. - Through this process, the efficiency of separating MuSCs was increased.	Yoshioka, Kitajima [29]

^1^ NS: not specified in the study. MuSCs, muscle stem cells; FACS, fluorescence-activated cell sorting; MACS, magnetic-activated cell sorting.

**Table 3 animals-14-00819-t003:** MuSC isolation using the density gradient centrifugation method.

Species (Muscle Types)	Detailed Method	Specific Features	Reference
Rat (extensor digitorum longus and tibialis anterior muscles)	- Used density: 25%, 35%, 40%, 55%, and 70% Percoll gradient - Centrifugation conditions: 1250× *g* for 20 min - Used fraction: 35%, 40%, and 55% fraction, 25/35%, 35/40%, 40/55%, and 55/70% interface fraction	- After culturing all fractions for 8 d, the overall frequency of the myogenic cells was lowest in the cultures of the 25/35% and 35% Percoll fractions and the highest in the cultures of the 55% fraction.	Kästner, Elias [49]
Rat (hind leg, extensor digitorum longus, and tibialis anterior muscles)	- Used density: 35%, 50%, and 70% Percoll gradient - Centrifugation conditions: 1250× *g* for 20 min - Used fraction: 50/70% interface fraction	-	Bischoff [50]
Rat (diaphragm, soleus, and tibialis anterior muscles)	- Used density: 20% and 60% Percoll gradient - Centrifugation conditions: 11,000× *g* for 5 min (SS-34 rotor of a Sorvall centrifuge) - Used fraction: 20/60% interface fraction	-	Dusterhöft, Yablonka-Reuveni and Pette [51]
Rat (hind limb muscles)	- Used density: 15%, 25%, 35%, 40%, 55%, and 70% Percoll gradient - Centrifugation conditions: 1250× *g* for 20 min - Used fraction: 15% fraction, 15/25%, 25/35%, 35/55%, and 55/70% interface fraction	- The 15/25% interface fraction had the highest percentage of stem cell antigen-1 cells used as a stem cell marker.	Che, Guo [33]
Rat (hind limb and back muscles)	- Used density: 27.5%, 35%, 40%, 55%, and 90% Percoll gradient - Centrifugation conditions: 1680× *g* for 20 min - Used fraction: 27.5/35%, 35/40%, and 40/55% interface fraction	- The expression of the satellite cell marker genes *MyoD* and *c-met* was the highest at the 40/55% interface fraction. - The expression of the fibroblast marker genes *FGF7* and *colla1* was the lowest at the 40/55% interface fraction.	Matsuyoshi, Akahoshi, Nakamura, Tatsumi and Mizunoya [52]
Pig (semimembranosus and longissimus dorsi muscle)	- Used density: 25%, 40%, and 70% Percoll gradient (condition 1), 25%, 40%, 50%, and 70% Percoll gradient (condition 2) - Centrifugation conditions: NS ^1^ - Used fraction: 40/70% interface fraction (condition 1), 40/50% and 50/70% interface fractions (condition 2)	- The 40/50% interface fraction had a high fusion rate and high myotube formation activity. - The 40/50% interface fraction demonstrated the highest myogenic differentiation potential.	Miersch, Stange [19]
Pig (semimembranosus muscles)	- Used density: 25%, 40%, and 90% Percoll gradient - Centrifugation conditions: 1800× *g* for 60 min, at 4 °C - Used fraction: 40/90% interface fraction	- The yield of the cells obtained through density gradient centrifugation was 4.1 × 10^6^ cells/g of muscle. - The concentration of the desmin-positive cells at the 40/50% interface fraction was >95%.	Mau, Oksbjerg [53]
Pig (longissimus and rhomboideus muscles)	- Used density: 20% Percoll gradient - Centrifugation conditions: 15,000× *g* for 8 min, at 4 °C - Used fraction: lower part of the 20% density	- The yield of cells obtained through density gradient centrifugation was 1.5 × 10^6^ cells/g of muscle.	Perruchot, Ecolan [54]
Pig (semitendinosus muscles)	- Used density: 20% and 60% Percoll gradient - Centrifugation conditions: 15,000× *g* for 5 min - Used fraction: 20/60% interface fraction	-	Mesires and Doumit [55]
Goat (longissimus dorsi muscles)	- Used density: 20%, 40%, and 90% Percoll gradient - Centrifugation conditions: 1800× *g* for 50 min - Used fraction: 40/90% interface fraction	-	Zhao, Chen [56]
Goat (rectus abdominis muscles)	- Used density: 20%, 40%, and 90% Percoll gradient - Centrifugation conditions: 1200× *g* for 90 min at 4 °C - Used fraction: 20/40% interface fraction	- Pre-plating was additionally performed before gradient centrifugation.	Tripathi, Ramani [57]

^1^ NS: not specified in the study.

**Table 4 animals-14-00819-t004:** Isolation of MuSCs using the FACS method.

Species (Types of Muscles)	Detailed Method	Specific Features	Reference
Mouse (tibialis anterior muscles)	- FACS buffer: 1% BSA in PBS - Positive/negative marker: Integrin α7^+^, CD34^+^/CD31^−^, CD45^−^, CD11^−^, Sca-1^−^ - Sorting laser: NS ^1^	-	Fu, Xiao [58]
Mouse (hind limb muscles)	- FACS buffer: 1% FBS in PBS - Positive/negative marker: SM/C2.6^+^/CD31^−^, CD45^−^, Sca-1^−^ - Sorting laser: NS	- The portion of PAX7^+^ and α7integrin^+^ cells in the MuSCs isolated using FACS was 94% and 96%, respectively	Urbani, Piccoli [59]
Mouse (triceps brachii, latissimus dorsi, and 7 other muscle types)	- FACS buffer: NS - Positive marker: calcein^+^, CD29^+^, CXCR4^+^, VCam1^+^, Integrin α7^+^, CD34^+^ - Negative marker: Sca-1^−^, CD31^−^, CD45^−^, CD11b^−^, Ter119^−^ - Sorting laser: NS	- MuSC fractions separated via FACS were divided into three types using different cell surface markers - Fraction 1; Calcein^+^CD29^+^CXCR4^+^, Fraction 2; Calcein^+^VCam1^+^, Fraction 3; Calcein^+^Integrin α7^+^CD34^+^ - The proportions of PAX7-expressing cells in fraction1, fraction 2, and fraction 3 were 90.3%, 90.4%, and 89.7%, respectively	Maesner, Almada [60]
Mouse (hind limb muscles)	- FACS buffer: 2% FBS in PBS - Positive/negative marker: Integrin α7^+^, CD34^+^/CD31^−^, CD45^−^, CD11b^−^, Sca-1^−^ - Sorting laser: 488, 633 nm, and UV	- Integrin α7^+^CD34^+^/CD31^−^CD45^−^CD11b^−^Sca-1^−^ satellite cells contain >90% of the PAX7^+^ satellite cells.	Pasut, Oleynik [61]
Pig (NS)	- FACS buffer: 1% BSA in PBS - Positive/negative marker: CD56^+^, CD29^+^/CD31^−^, CD45^−^ - Sorting laser: 488, 561, and 640 nm	- Among the CD31^−^CD45^−^CD56^+^CD29^+^ cells isolated using FACS, the proportion of cells expressing PAX7 was 94%.	Ding, Wang [44]
Pig (NS)	- FACS buffer: 1% BSA in PBS - Positive/negative marker: CD56^+^, CD29^+^/CD31^−^, CD45^−^ - Sorting laser: 488, 561, and 640 nm	- The yield of cells obtained through FACS separation was 5.3 × 10^4^ cells/g of muscle. - The PAX7 expression population of the MuSCs obtained through FACS sorting was 87%.	Zhu, Wu [62]
Pig (NS)	- FACS buffer: 1% BSA in PBS - Positive/negative marker: CD56^+^, CD29^+^/CD31^−^, CD45^−^ - Sorting laser: 405, 488, and 640 nm	-	Guan, Pan [10]
Pig (NS)	- FACS buffer: NS - Positive/negative marker: CD56^+^, CD29^+^/CD31^−^, CD45^−^ - Sorting laser: NS	- The PAX7 expression population of the MuSCs obtained through FACS sorting was 95%.	Li, Wang [30]
Cattle (NS)	- FACS buffer: 1% BSA in PBS - Positive/negative marker: CD56^+^, CD29^+^/CD31^−^, CD45^−^ - Sorting laser: 405, 488, and 640 nm	- The ratio of PAX7^+^ cells in the MuSCs isolated using FACS was 92%. - The proportions of MYOD-, desmin-, and MYF5-expressing cells were 98%, 98%, and 97%, respectively.	Ding, Swennen [63]
Human (NS)	- FACS buffer: NS - Positive/negative marker: ITGB1^+^/CD31^−^, CD34^−^, CD45^−^ - Sorting laser: 405, 488, and 633 nm	- The PAX7 expression population of MuSCs obtained through FACS sorting was 96%. - No PAX7 expression was observed in the CD34^−^, CD31^−^, and CD45-expressing cells.	Charville, Cheung [64]

NS ^1^: not specified in the study. MuSCs, muscle stem cells; FACS, fluorescence-activated cell sorting; MACS, magnetic-activated cell sorting; BSA, bovine serum albumin; PBS, phosphate-buffered saline; FBS, fetal bovine serum.

**Table 5 animals-14-00819-t005:** MuSC isolation using the MACS method.

Species (Types of Muscles)	Detailed Method	Specific Features	Reference
Mouse (posterior limb muscles)	- MACS buffer: 0.5% BSA and 2 mM EDTA in PBS - Positive/negative marker: α7 integrin^+^/CD31^−^, CD45^−^, Sca1^−^, CD11b^−^ - Sorting method: (1) The cells were incubated with CD31, CD45, Sca1, and CD11b primary antibodies. (2) After centrifugation, the streptavidin microbead mixture was mixed with cells and incubated. (3) The cells were passed through a column and placed in the MACS separator. (4) Non-labeled cells were obtained and incubated with the α7 integrin primary antibody. (5) After centrifugation, the streptavidin microbead mixture was mixed with cells and incubated. (6) The cells were passed through the column and placed in the MACS separator. (7) Non-labeled cells were washed through. (8) After removing the magnetic field, α7 integrin^+^ fraction cells were eluted.	- To increase the separation efficiency of MuSCs, MACS separation was performed twice using positive and negative markers. - Extremely effective in removing fibroblast contamination from culture mixtures.	Sincennes, Wang [16]
Mouse (tibialis anterior, gastrocnemius, and quadriceps muscles)	- MACS buffer: NS ^1^ - Positive/negative marker: α7 integrin^+^/CD31^−^, CD45^−^, Sca1^−^ - Sorting method: (1) The cells were incubated with CD31, CD45, Sca1, and α7 integrin primary antibodies. (2) After centrifugation, the anti-PE magnetic beads were mixed with cells and incubated. (3) The labeled cells were passed through the column and placed in the MACS separator. (4) Non-labeled cells were obtained and incubated using anti-mouse IgG magnetic beads. (5) The cells were passed through the column and placed in the MACS separator. (6) Non-labeled cells were washed through. (7) After removing the magnetic field, α7 integrin^+^ fraction cells were eluted.	- To increase the separation efficiency of MuSCs, MACS separation was performed twice using positive and negative markers. - >90% of the isolated cells expressed PAX7.	Motohashi, Asakura [66]
Mouse (hind limb muscles)	- MACS buffer: 2% FCS in DMEM - Positive/negative marker: CD44^+^, CD90^+^, α7 integrin^+^/NU ^2^ - Sorting method: (1) The cells were reacted with magnetic microbeads conjugated to the CD90, CD44, and α7 integrin primary antibodies. (2) The labeled cells were passed through the column and placed in the MACS separator. (3) Non-labeled cells were washed through. (4) After removing the magnetic field, CD44^+^, CD90^+^, α7 integrin^+^ fraction cells were eluted.	-	Elashry, Kinde [23]
Pig (biceps femoris muscles)	- MACS buffer: NS - Positive/negative marker: CD29^+^/NU - Sorting method: (1) The cells were reacted with an anti-CD29 antibody and anti-mouse IgG microbeads. (2) The labeled cells were passed through a column and placed in the MACS separator. (3) Non-labeled cells were washed through. (4) After removing the magnetic field, the CD29^+^ fraction cells were eluted.	- Before MACS isolation, the proportion of non-myogenic cells in the cultured cells reached 30%. - After MACS isolation, the ratio of CD56^+^/CD29^+^ MuSCs was 91.5%.	Choi, Kim [14]
Cattle (biceps femoris muscles)	- MACS buffer: NS - Positive/negative marker: CD29^+^/NU - Sorting method: (1) The cells were reacted with an anti-CD29 antibody and anti-mouse IgG microbeads. (2) The labeled cells were passed through the column and placed in the MACS separator. (3) Non-labeled cells were washed through. (4) After removing the magnetic field, the CD29^+^ fraction cells were eluted.	-	Kim, Ko [67]
Human (NS)	- MACS buffer: 1% BSA and EDTA in PBS - Positive/negative marker: CD56^+^/NU - Sorting method: (1) The cells were reacted with magnetic microbeads conjugated to a CD56 primary antibody. (2) The labeled cells were passed through the column and placed in the MACS separator. (3) Non-labeled cells were washed through. (4) After removing the magnetic field, the CD56^+^ fraction cells were eluted.	- After MACS isolation, the myogenic cell content of the CD56^+^ cell fraction was >95%.	Agley, Rowlerson [68]
Human (NS)	- MACS buffer: BSA, EDTA, and 0.09% azide in PBS - Positive/negative marker: CD56^+^/NU - Sorting method: (1) The cells were magnetically labeled with super-paramagnetic antigen (CD56)-specific MACS MicroBeads. (2) The labeled cells were passed through the column and placed in the MACS separator. (3) Non-labeled cells were washed through. (4) After removing the magnetic field, the CD56^+^ fraction cells were eluted.	- The proportion of the myogenic cells (desmin-positive cells) in the heterogeneous mixture before MACS separation was 19%. - After MACS isolation, the proportion of myogenic cells increased to 85%.	Brady, Lewis [69]
Human (NS)	- MACS buffer: BSA, EDTA, and 0.09% azide in PBS - Positive/negative marker: CD56^+^/NU - Sorting method: (1) The cells were magnetically labeled with super-paramagnetic antigen (CD56)-specific MACS MicroBeads. (2) The labeled cells were passed through the column and placed in the MACS separator. (3) Non-labeled cells were washed through. (4) After removing the magnetic field, the CD56^+^ fraction cells were eluted.	-	Martin, Passey [70]
Humans (NS)	- MACS buffer: 0.2% BSA in PBS - Positive/negative marker: CD56^+^/NU - Sorting method: (1) The cells were reacted with magnetic microbeads conjugated to a CD56 primary antibody. (2) The labeled cells were passed through the column and placed in the MACS separator. (3) Non-labeled cells were washed through. (4) After removing the magnetic field, the CD56^+^ fraction cells were eluted.	- The yield of cells obtained through MACS separation was 2 × 10^5^ cells/g of muscle.	Wang, Broer [71]

NS ^1^: not specified in the study. NU ^2^: not used in this study. MACS, magnetic-activated cell sorting; MuSC, muscle stem cell; BSA, bovine serum albumin; EDTA, ethylenediamine tetraacetic acid; PBS, phosphate-buffered saline.

## Data Availability

Data are contained within the article.

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
