# Peer review of "Methods to Isolate Muscle Stem Cells for Cell-Based Cultured Meat Production: A Review"

_animals, 2024, doi:10.3390/ani14050819_

Round 1

Reviewer 1 Report

Comments and Suggestions for Authors

Dear Authors,

This review describes and compares different methodological approaches to isolate the muscle stem cells out of muscles samples. These cells might possess higher muscle differentiation capabilities compared to the other cell lines, that could be used in cultured meat development.

I found the review methodologically very instructive and informative. It thoroughly describes the various isolation approaches currently known. In my opinion, both the concept of the review and its structure are well thought out and understandable. The manuscript is well written. Therefore, I suggest accepting the manuscript after correcting or clarifying some minor statements.

MINOR COMMENTS:

Table 1

I think Table 1 would fit better in section 5. Conclusion and future directions/recommendations. I suggest adapting/reorganising the sentences accordingly.

Table 2-5

I wonder if there is a specific reason why some researchers do not specify muscle type in their studies? I think that an anatomical part of the muscles (the origin from which a sample comes) is important both for the methodological approach and for a better comparison in general and specifically in this review.

4. Research to increase MuSCs separation efficiency

Line 301

Please leave out the word "the".

Line 303-308 (reference 52)

Did the authors also show a better result in the purity of MuSCs with the combination of methods used, apart from being able to remove a significant proportion of fibroblasts? If available, I suggest adding this information.

Line 308-309 (reference 69)

The same remark. Did the authors also show a better result in terms of MuSCs purity with the combination of methods used (MACS and FACS)? If available, I suggest adding this information.

Line 316-318 (They reported the 316 myogenic cell ratio to be substantially higher than that of the MuSCs derived and isolated 317 using the ICT method and using MACS, respectively)

Please rewrite this sentence, because it is quite puzzling to understand.

5. Conclusions and future directions/recommendations

I am aware that there is room for optimisation and improvement in each methodological approach described and that there are many factors that need to be considered when choosing the "right" method. However, regardless of the different species, muscle types, and financial aspects (equipment, reagents, maintenance), could you point out a favorable, generalised method for the isolation of this specific MuSC cell line?

Comments on the Quality of English Language

I have no particular comment.

Author Response

March 4, 2024

Animals  

Manuscript ID: animals-2881299

Title: Methods to isolate muscle stem cells for cell-based cultured meat production: a review

Dear Editor:

We wish to thank you and the reviewers for the constructive comments regarding our manuscript. The manuscript has greatly benefited from these helpful suggestions. Each comment has been addressed in detail, as shown below, and the corrections are indicated in red text in the revised manuscript for your ease of review. We look forward to working with you and the reviewers to move this manuscript closer to publication in Animals.

[Reviewer 1’s comments]

This review describes and compares different methodological approaches to isolate the muscle stem cells out of muscles samples. These cells might possess higher muscle differentiation capabilities compared to the other cell lines, that could be used in cultured meat development. I found the review methodologically very instructive and informative. It thoroughly describes the various isolation approaches currently known. In my opinion, both the concept of the review and its structure are well thought out and understandable. The manuscript is well written. Therefore, I suggest accepting the manuscript after correcting or clarifying some minor statements.

Comment 1: Table 1, I think Table 1 would fit better in section 5. Conclusion and future directions/recommendations. I suggest adapting/reorganising the sentences accordingly.

Response 1: Thank you for your valuable comment. However, when writing this review paper, we wanted to provide in [3. MuSCs isolation methods] part a rough introduction to the four types of MuSC isolation methods by explaining their simple principles, advantages, and disadvantages. Next, using subheadings 3.1.~3.4., we tried to focus on detailed explanations of each method and actual use cases accordingly. Therefore, in the case of Table 1, the principles, advantages, and disadvantages of each method are explained in lines 102-141 corresponding to the introduction of [3. MuSCs isolation methods], so we think it is more reasonable for Table 1 to be inserted at this location.

Comment 2: Table 2-5, I wonder if there is a specific reason why some researchers do not specify muscle type in their studies? I think that an anatomical part of the muscles (the origin from which a sample comes) is important both for the methodological approach and for a better comparison in general and specifically in this review.

Response 2: Like the reviewer, we were disappointed that some references did not specifically mention the origin of the muscle sample. However, even after carefully reviewing the references, there was no specific mention of muscle type. Nevertheless, we thought that these references could provide some useful information to this review paper that organizes and summarizes the separation method for isolating muscle stem cells, so we added those references to each table.

Comment 3: Line 301, Please leave out the word "the".

Response 3: We left out the word "the". (Line 306)

Comment 4: Line 303-308 (reference 52), Did the authors also show a better result in the purity of MuSCs with the combination of methods used, apart from being able to remove a significant proportion of fibroblasts? If available, I suggest adding this information.

Response 4: The yield values obtained by combining pre-plating and density gradient centrifugation and the yield values obtained when each method was performed independently, which can be found in the reference, are presented in the manuscript. (Line 313-316)

Comment 5: Line 308-309 (reference 69), The same remark. Did the authors also show a better result in terms of MuSCs purity with the combination of methods used (MACS and FACS)? If available, I suggest adding this information.

Response 5: Unfortunately, the reference (69) did not mention the exact purity values obtained when using MACS and FACS in combination. However, it is mentioned that the yield of MuSCs was increased through the combined method.

Comment 6: Line 316-318 (They reported the myogenic cell ratio to be substantially higher than that of the MuSCs derived and isolated using the ICT method and using MACS, respectively), Please rewrite this sentence, because it is quite puzzling to understand.

Response 6: The sentence has been rewritten to help readers understand. (Line 324-326)

Comment 7: 5. Conclusions and future directions/recommendations, I am aware that there is room for optimisation and improvement in each methodological approach described and that there are many factors that need to be considered when choosing the "right" method. However, regardless of the different species, muscle types, and financial aspects (equipment, reagents, maintenance), could you point out a favorable, generalised method for the isolation of this specific MuSC cell line?

Response 7: While reviewing papers related to various MuSC isolation methods, we found that no single isolation method could be recommended as the best. As can be seen in Table 1 and the manuscript, each isolation method has advantages and disadvantages, and it was also confirmed that the corresponding advantages and disadvantages complement each other. In addition, when we confirmed that the extraction yield and purity of MuSCs vary depending on how the extraction conditions are set even in one method, we realized in this study that research is needed to select and optimize the method needed for each study. This was the conclusion we came to while writing a review paper. In addition, it was said that it would be possible to isolate MuSCs with higher yield and purity by incorporating new technology rather than using only one method. Therefore, rather than choosing one method in the conclusion, we would like to conclude the review paper by arguing the above points to the authors.

Once again, we thank the reviewers for their meticulous review of our work. We believe that these suggestions have strengthened the impact of the paper.

Thank you for your consideration. I look forward to hearing from you.

Reviewer 2 Report

Comments and Suggestions for Authors

L 41 - technical, ethical and regulatory challenges..- could you please give some examples

Is it important that for isolation of MuSc we use slaughtered animals? Could you please discuss more in detail this fact. If yes, it is contradictory to L 38

Conclusion - please conclude, do not repeat what was your aim, etc., just conclude in short (l 232-327 please delete)

Please correct your references in the manuscript (e.g., Perruchot et al...L 190; Kastner, et al..L 208; Chee et al... L 209, etc.)

Author Response

March 4, 2024

Animals  

Manuscript ID: animals-2881299

Title: Methods to isolate muscle stem cells for cell-based cultured meat production: a review

Dear Editor:

We wish to thank you and the reviewers for the constructive comments regarding our manuscript. The manuscript has greatly benefited from these helpful suggestions. Each comment has been addressed in detail, as shown below, and the corrections are indicated in red text in the revised manuscript for your ease of review. We look forward to working with you and the reviewers to move this manuscript closer to publication in Animals.

[Reviewer 2’s comments]

Comment 1: L 41 - technical, ethical and regulatory challenges..- could you please give some examples

Response 1: We have added simple examples of the technical, ethical, and regulatory challenges that must be overcome for cultured meat to be commercialized in this sentence. (Line 40-44)

Comment 2: Is it important that for isolation of MuSc we use slaughtered animals? Could you please discuss more in detail this fact. If yes, it is contradictory to L 38.

Response 2: There are two ways to obtain MuSCs: one is to obtain them from slaughtered meat, and the other is to obtain them from live animals through biopsy. If MuSCs are obtained through the biopsy method alone, cultured meat can be produced without slaughter, and even if slaughter is performed, large amounts of cultured meat can be produced even with a small amount of cells collected, which is expected to dramatically reduce the amount of animal slaughter. Therefore, we think it does not contradict Line 38 as pointed out by the reviewer.

Comment 3: Conclusion - please conclude, do not repeat what was your aim, etc., just conclude in short (L 323-327 please delete)

Response 3: In order to write a concise conclusion, the sentence you pointed out was deleted.

Comment 4: Please correct your references in the manuscript (e.g., Perruchot et al...L 190; Kastner, et al..L 208; Chee et al... L 209, etc.)

Response 4: We have reviewed the reference format of the manuscript as a whole, including the references you mentioned.

Once again, we thank the reviewers for their meticulous review of our work. We believe that these suggestions have strengthened the impact of the paper.

Thank you for your consideration. I look forward to hearing from you.

Reviewer 3 Report

Comments and Suggestions for Authors

The paper, entitled "Methods to isolate muscle stem cells for cell-based cultured meat production: a review" by Jae Hoon Lee et al., reviewed four major techniques for isolating MuSCs in cultured meat production. This review outlines pre-plating, density gradient centrifugation, FACS, and MACS methods for isolating MuSCs from tissue, and compares the advantages and disadvantages of each approach. Additionally, the authors emphasized the primary concept behind these methods, aimed at enhancing the efficiency of MuSCs. The last but not least, the authors recommended that the optimal utilization of these methods for isolating MuSCs will depend on the specific livestock species and muscle source. The manuscript is well-written. The reviewer has only one minor concern as follows:

1. In the Introduction section, could the authors provide further description regarding the potential impacts of cultured meat on the human protein supply chain and the real meat market, as well as discuss potential conditions that could facilitate the transition of cultured meat to industrial-scale production?

2. For cultured meat production, MuSCs may be the major cell type used for culture, and the authors described various benefits of using MuSCs. However, the authors should also outline a few disadvantages of using MuSCs for cultured meat production and current solutions to address those disadvantages.

Comments on the Quality of English Language

I am comfortable with the quality of the author's English language.

Author Response

March 4, 2024

Animals  

Manuscript ID: animals-2881299

Title: Methods to isolate muscle stem cells for cell-based cultured meat production: a review

Dear Editor:

We wish to thank you and the reviewers for the constructive comments regarding our manuscript. The manuscript has greatly benefited from these helpful suggestions. Each comment has been addressed in detail, as shown below, and the corrections are indicated in red text in the revised manuscript for your ease of review. We look forward to working with you and the reviewers to move this manuscript closer to publication in Animals.

[Reviewer 3’s comments]

The paper, entitled "Methods to isolate muscle stem cells for cell-based cultured meat production: a review" by Jae Hoon Lee et al., reviewed four major techniques for isolating MuSCs in cultured meat production. This review outlines pre-plating, density gradient centrifugation, FACS, and MACS methods for isolating MuSCs from tissue, and compares the advantages and disadvantages of each approach. Additionally, the authors emphasized the primary concept behind these methods, aimed at enhancing the efficiency of MuSCs. The last but not least, the authors recommended that the optimal utilization of these methods for isolating MuSCs will depend on the specific livestock species and muscle source. The manuscript is well-written. The reviewer has only one minor concern as follows:

Comment 1: In the Introduction section, could the authors provide further description regarding the potential impacts of cultured meat on the human protein supply chain and the real meat market, as well as discuss potential conditions that could facilitate the transition of cultured meat to industrial-scale production?

Response 1: The future prospects and mass production of cultured meat as a protein source pointed out by the reviewer have been newly added to the Introduction part. (Line 51-55)

Comment 2: For cultured meat production, MuSCs may be the major cell type used for culture, and the authors described various benefits of using MuSCs. However, the authors should also outline a few disadvantages of using MuSCs for cultured meat production and current solutions to address those disadvantages.

Response 2: A known disadvantage of MuSCs is that their proliferation capacity and stemness decrease with subculture. Nevertheless, compared to other cell types (embryonic stem cell lines, induced pluripotent cell lines, and naturally immortalized cell lines), its differentiation ability is far superior. It is expected that MuSCs will occupy a major position in the future cultured meat market as they can provide a texture similar to real meat. Therefore, we wrote a review paper summarizing methods for efficiently isolating MuSCs. In the conclusion section, we wrote about these disadvantages, advantages, and future prospects. However, since our review paper does not aim to overcome the disadvantages of MuSCs, we did not specifically mention solutions to overcome them. (Line 339-343)

Once again, we thank the reviewers for their meticulous review of our work. We believe that these suggestions have strengthened the impact of the paper.

Thank you for your consideration. I look forward to hearing from you.